# On the Structural and Molecular Properties of PEO and PEO-PPG Functionalized Chitosan Nanoparticles for Drug Delivery

**DOI:** 10.3390/bioengineering11040372

**Published:** 2024-04-12

**Authors:** Rejeena Jha, Hyrum Harlow, Mourad Benamara, Robert A. Mayanovic

**Affiliations:** 1Department of Physics, Astronomy, and Materials Science, Missouri State University, Springfield, MO 65804, USA; rj23s@login.missouristate.edu (R.J.);; 2University of Arkansas Nano-Bio Materials Characterization Facility, University of Arkansas, Fayetteville, AR 72701, USA

**Keywords:** chitosan nanoparticles, structure properties, polymer functionalization, X-ray diffraction, transmission electron microscopy, Fourier transform infrared spectroscopy

## Abstract

Chitosan nanoparticles (CS-NPs) are currently under investigation for a wide range of applications in nanomedicine. We investigated the structural, morphological, and molecular properties of CS-NPs synthesized via ionic gelation and designed specifically for drug delivery. The CS-NPs were prepared at concentrations ranging from 0.25 to 1.0% *w*/*v*. The 1.0% *w*/*v* CS-NPs were also functionalized with polyethylene oxide (PEO) alone and with a diblock copolymer of PEO and polypropylene glycol (PPG). The average nanoparticle size determined from TEM imaging is in the 11.3 to 14.8 nm range. The XRD and TEM analyses reveal a semi-crystalline structure with a degree of crystallinity dependent upon the nature of CS-NP functionalization. Functionalizing with PEO had no effect, whereas functionalizing with PEO-PPG resulted in a significant increase in the crystallinity of the 1.0% *w*/*v* CS-NPs. Additionally, the CS/TPP concentration (CS:TPP fixed at a 1:1 ratio) did not impact the degree of crystallinity of the CS-NPs. FTIR analysis confirmed the incorporation of TPP with CS and an increase in hydrogen bonding in more crystalline CS-NPs.

## 1. Introduction

Chitosan nanoparticles are currently under vigorous investigation as suitable agents for in vivo delivery of drugs and genetic material [1,2,3,4,5,6]. Current research has shown that their small size and selective functionalization enable chitosan nanoparticles (CS-NPs) to enter human tissue in a targeted manner [2,3,4,5,6,7,8]. This targeted modality of treatment has distinct advantages over many conventional treatment approaches, such as chemotherapy in the treatment of cancer. Furthermore, the fact that CS-NPs are derived from chitosan, which is a natural biodegradable polysaccharide material, provides a distinct biocompatibility advantage over inorganic nanoparticles. Understanding the structural characteristics of polymer-modified CS-NPs is critical for enhancing their drug delivery properties.

Chitosan (CS) is a polysaccharide derived from the N-deacetylation of chitin. The molecular structure of chitosan is a linear copolymer consisting of glucosamine and N-acetyl-glucosamine units: the repeatability of the units is determined by the degree of deacetylation (DD) [9,10]. The level of deacetylation directly affects the efficacy of CS in a wide range of applications [11,12,13,14]. The amount of protonated NH_2_ groups determines the solubility of chitosan in aqueous media [9]. Chitosan is soluble in acidic solutions at relatively low DD values. Therefore, assuming all other variables remain constant, the DD level directly affects the solubility, crystal structure, and size of glucosamine units in chitosan [9]. As a result, the degree of crystallinity and crystal size potentially plays a significant role in the biochemical, nanomedicinal, and pharmacological applications of CS-NPs [15,16].

The methodologies employed to synthesize CS-NPs include ionic gelation, microemulsion, emulsion-based solvent evaporation, and emulsification solvent diffusion [17,18,19]. The particle size, crystallinity, and surface charge of the CS-NPs are principally determined by the molecular weight, the amount, and the DD of CS. Chitosan is a semi-crystalline biopolymer consisting of both a crystalline and an amorphous component [20]. Chitosan can exhibit diverse crystalline structural forms based on the various raw material sources, α-chitin, β-chitin, and γ-chitin, in a mixed arrangement of the two allomorphs [21]. In contrast to β-chitin, which exhibits a monoclinic P2_1_ symmetry with parallel displacement and decreased intersheet interaction, α-chitin maintains the orthorhombic P2_1_2_1_2_1_ symmetry with antiparallel displacement of the polymeric chains [22]. The XRD patterns of the chitin allomorphs generally exhibit two high-intensity diffraction peaks occurring within the 8–11° and 19–21° ranges in 2θ, which coincide with the (110) and (120) reflections, respectively. Conversely, secondary peaks are primarily observed in XRD patterns from α-chitin [21,22,23]. Depending upon the procedures used in processing, chitosan can occur in hydrated form. Fachinatto et al. made a detailed structure study using XRD and NMR of CS showing that increasing the DD level increases the degree of crystallinity [23]. In addition, this study showed that the degree of crystallinity is minimally impacted by molecular weight, particularly at lower DD levels. This was confirmed by Savitri et al. [24] showing that decreasing the molar mass with DA > 20% has a direct bearing on the degree of crystallinity of the CS.

In this study, CS-NPs were functionalized using polyethylene oxide (PEO) and a diblock copolymer of PEO and polypropylene glycol (PPG). Polymer-based functionalization enhances solubility, hemocompatibility, bodily fluid stability, and drug delivery capacity while reducing the cytotoxicity of CS-NPs [25]. Despite the importance of such modifications, studies on the structural characteristics of polymer-modified CS-NPs are scarce. However, fully understanding the structure of polymer-modified CS-NPs is paramount for advancing drug delivery technologies. Structural characteristics, including the degree of crystallinity, profoundly influence the physicochemical properties of CS-based drug delivery systems [26]. These properties include their stability, solubility, degradation kinetics, and interactions with both drugs and biological tissues [27]. By leveraging this structural knowledge, CS-NP-based drug delivery systems can be tailored to meet specific therapeutic requirements, thus ensuring optimal drug release kinetics, improving targeting capabilities, and enhancing therapeutic efficacy [25]. Moreover, understanding the structural features governing biocompatibility and safety enables the development of safer formulations [28]. In essence, structural studies serve as the cornerstone for the design, development, and regulatory approval of chitosan-based drug delivery systems, fostering innovation and improving therapeutic outcomes [2].

We used the ionic gelation method to synthesize CS-NPs of varying concentrations and ones functionalized with PEO or the PEO-PPG diblock copolymer. The resulting CS-NPs were fully characterized using X-ray diffraction (XRD), transmission electron microscopy (TEM), and Fourier transmission infrared spectroscopy (FTIR) to determine the structure and bonding properties. The primary focus of this study, which is reported herein, was to investigate whether the structural characteristics of the CS-NPs were affected by concentration and by the type of surface functionalization, and if so, in which manner. We show for the first time that using XRD and FTIR characterization along with a detailed high-resolution TEM analysis enables the discovery of structural characteristics that may provide pathways toward optimization of the drug delivery capacity of surface functionalized CS-NPs.

## 2. Materials and Methods

### 2.1. Preparation of CS Nanoparticles

The polyethylene oxide (PEO) of molecular weight (MW) 300,000, polypropylene glycol (PPG) of MW of 4000, sodium tripolyphosphate (TPP), and 85% deacetylated chitosan (CS) that were used in this study were purchased from Thermo Fisher Scientific Chemicals, Inc. (Ward Hill, MA 01835-8099,USA). The CS-NPs were synthesized using the ionic gelation method [29]. Here, an aqueous solution of CS is mixed with an aqueous solution of sodium tripolyphosphate (TPP). The opposing charges of CS and TPP cause inter- and intra-molecular cross-linking, resulting in the formation of chitosan nanoparticles.

Bulk chitosan and TPP (in equal proportions) were each prepared in concentrations of 0.25%, 0.5%, and 1.0% *w*/*v*. CS was dissolved in high-purity liquid chromatography (HPLC) water, containing glacial acetic acid, using a magnetic stirrer set to stir vigorously for approximately 2–3 h at room temperature. In each instance, the acetic acid concentration was 1.75 times that of CS [29]. Similarly, TPP was dissolved in a separate beaker containing HPLC water using a magnetic stirrer set to stir vigorously for 2–3 min at room temperature. Subsequently, the TPP was added to the CS solution under magnetic stirring dropwise using a pipette. During the synthesis process, three different stages were identified as TPP solution was progressively added to the CS solution: clear, opalescent, and aggregated solution of CS-NPs. At low CS concentrations, the opalescence observed was very faint which became more prominent with higher CS concentration. Subsequently, each aggregated sample was centrifuged using 8:1 ethanol–toluene solution 5 times to remove any impurities. The centrifuged sample was then dried and ground using mortar and pestle into the powdered form for characterization.

Functionalized CS-NPs were formed spontaneously upon the incorporation of 8 mL of 1% *w*/*v* of TPP and 50 mg of PEO or PEO-PPG in 20 mL of 1% *w*/*v* CS solution. The surfactant and diblock copolymer concentration was 1.75 times higher than the CS concentration. The synthesis and harvesting procedures were the same as outlined above for the synthesis of the non-functionalized CS-NPs. The samples used in this study are shown in Table 1.

### 2.2. Characterization of Chitosan Nanoparticles

X-ray diffraction, transmission electron microscopy, and Fourier transform infrared spectroscopy were employed for the structural and morphological characterization of the samples. XRD is a highly useful characterization technique for investigations of the degree of crystallinity and structural characteristics of CS-NPs [1,21,23]. For samples possessing a high degree of crystallinity, XRD may enable the determination of their mean particle size using the Scherrer equation. The CS-NP samples were fixed on glass slides using ethanol for the XRD measurements. The characterization was made using a Bruker D8 Discover diffractometer with Cu Kα (λ = 1.5406 Å) radiation. During the data collection, the same geometry and measurement parameters were used to ensure a systematic comparison of the XRD spectra. Care was taken for the proper alignment of the sample in the diffractometer for optimal data collection. Up to 8 individual XRD spectra were collected and averaged for measurements from each sample.

The size, morphology, and degree of crystallinity of nanoparticles can be very effectively investigated using low- and high-resolution TEM [1]. The TEM image of the nanoparticles is formed from the interference of their transmitted and diffracted electron beams. For TEM imaging, the samples were dissolved in hexane and placed on a carbon TEM grid. The TEM and high-resolution transmission electron microscopy (HRTEM) imaging of the samples was performed on an FEI Titan 80−300 instrument at the University of Arkansas Nano-Bio Materials Characterization Facility, with the field emission gun set at 300 keV. The extent of sample crystallinity, which was determined using fast Fourier transform (FFT) analysis, and measurement of CS-NP size were made using imageJ 1.52 software.

FTIR spectroscopy is a powerful technique used for the determination of the vibrational/molecular properties of polymer-based materials, including CS-NPs. The absorption of infrared light at frequencies characteristic of excitation of specific molecular vibrational modes is utilized to infer the polymeric structure and nature of macromolecular complexes in samples. The FTIR measurements were made on the CS-NPs samples using a Bruker Alpha II spectrometer. The Alpha II spectrometer is equipped with a Diamond Crystal ATR (attenuated total internal reflectance) accessory. The FTIR spectra were measured from our CS-NP samples in powder form.

## 3. Results and Discussion

### 3.1. XRD and TEM Analyses

Figure 1a shows the XRD data measured from bulk CS, the 1% *w*/*v* non-functionalized CS-NPs, and CS-NPs functionalized with PEO and with PEO-PPG. The XRD spectra measured from the CS-NPs of varying CS concentrations are shown in Figure 1b. As shown in Figure 1a, the XRD pattern measured from bulk CS consists of two predominant peaks, one occurring at 10.1°, with a planar d spacing of 0.875 nm, and a wider peak occurring at 19.9° in 2θ stemming from a planar d spacing of 0.446 nm. As mentioned above, peaks occurring in the 8–11° and 19–21° ranges are typically attributed to reflections from the (020) and (110) planes in the crystalline CS structure, respectively. The variability in the reported positions is typically attributed to the source of chitin used in the preparation of the chitosan [21]. The d spacings associated with the (020) and (110) reflections and the lack of additional notable reflections that occur for the α allomorph makes our bulk CS sample most likely the β allomorph of chitosan [21,22,23]. Interestingly, the XRD pattern measured from the bulk CS sample also shows two lower-intensity peaks, one at 14.9° and a smaller one at 29.2°. As seen in Figure 1a,b, the diffraction patterns measured from our CS-NP samples are characterized by a convolution of a very broad peak-like feature centered near 23° and broad diffraction peaks of smaller intensities. Our CS-NPs exhibit a reduced degree of crystallinity in comparison to that of bulk CS, which agrees with previous studies [30,31]. The extensive peak-like feature centered near 23° is indicative of a significant amorphous contribution to the overall structure of the CS-NPs: the smaller-intensity diffraction peaks are indicative of a crystalline contribution. The (110) peak is seen to be shifted in the XRD patterns measured from the CS-NPs to 18.6°, indicating an increased level of hydration in comparison to that of the bulk CS. This is consistent with the use of the aqueous solvent in the ionic gelation synthesis of the CS-NPs. Additionally, a lower intensity peak is found to occur in the 11.7–11.9° range for all CS-NPs, as shown in Figure 1. This peak may coincide with the crystalline plane reflections responsible for the 14.9° peak of bulk CS but shifted due to hydration. There is also a prominent shoulder observed near 30° in all XRD patterns measured from the CS-NPs, which is most likely correlated with the 29.2° peak of bulk CS. We are not aware of any previous report on XRD peaks or features located near 12° and 30° for CS-NPs. This requires further study to determine the crystalline structure characteristics that give rise to these diffraction features for CS-NPs.

Figure 1a clearly indicates that the functionalization with PEO has a negligible impact on the degree of crystallinity of the CS-NPs. This is in contradiction with the previous findings indicating that functionalization with PEO increases the crystallinity of CS-NPs [32]. This study purports that the formation of hydrogen bonds between the amine group from chitosan and PEO (via polyether O) increases chain entanglement, leading to increased crystallinity. We do not find evidence for such action from the PEO functionalization of our CS-NPs. However, we find that the addition of PEO-PPG has a significant impact on increasing the degree of crystallinity of the 1.0% *w*/*v* CS-NPs. We conjecture that this is due to the self-organization of the PEO-PPG block copolymer into ordered structures, such as micelles, in aqueous solutions. Mortensen et al. determined that the polystyrene (PS)-PEO diblock copolymer forms spherical micelles at up to 20% concentration in aqueous solutions [33]. The PEO-PPG block copolymer may first envelop the embryonic phase of the CS-NPs allowing for more controlled diffusive motion and greater organization of CS polymeric strands into crystalline units. Nevertheless, this hypothesis needs to be tested with additional structural studies of CS-NP formation in PEO-PPG-bearing aqueous solution using other techniques (e.g., small-angle neutron scattering). Figure 1b shows that the XRD patterns measured from the 0.25–1.0% *w*/*v* CS-NP samples (where CS:TPP ratio is fixed at 1:1) are qualitatively similar in appearance. This indicates that the CS/TPP concentration has a negligible effect on the degree of crystallinity of the CS nanoparticles. A small peak is evident in the XRD spectra of the non-functionalized CS-NP samples in the 8.6–8.9° range. This feature is consistent with a weak reflection from the chitosan (020) planes.

The morphology and size of CS-NPs were studied using TEM imaging whereas the structural aspects were studied using HRTEM. The CS-NPs are generally pseudo-spherical and prismatic shaped. As shown in Table 1, the size of the CS-NPs was found to be similar for all samples and in the 11–15 nm range. This is attributed to the fixed ratio of CS:TPP (1:1) used in the ionic gelation of the CS-NP samples. The semi-crystalline nature of the samples was verified using the fast Fourier transform (FFT) analysis of HRTEM images, which are shown in Figure 2 and Appendix A. Notably, our use of a detailed HRTEM analysis has enabled the discovery of novel morphological and structural features of functionalized CS-NPs. We observe a variability in the extent of crystallinity exhibited by the CS-NPs: some are predominantly amorphous with negligible crystallinity, whereas others (generally prismatic shaped) show varying degrees of crystallinity. The amorphous nature is partially exhibited in the FFTs by the wide and diffuse ring feature (hexane residue provides an additional contribution), whereas the spot-like pattern is indicative of reflections from specific crystalline planes. The FFTs of select nanoparticles in Figure 2c,f are consistent with a lower degree of crystallinity exhibited by the CNP-1% sample compared to that of the CNP-PEPG sample, respectively.

### 3.2. FTIR Spectroscopy Analysis

FTIR spectra of bulk CS, CNP-1%, CNP-PEO, and CNP-PEPG samples are shown in Figure 3. The broad peak in the ~3000–3500 cm^−1^ range is attributed to N-H and O-H stretching vibrations within the CS polymer chains [34,35]. This peak is more prominent for CNP-PEPG, indicating more enhanced hydrogen bonding compared to that in CNP-PEO and CNP-1%. The peak at 2861 cm^−1^ is due to the C-H asymmetric vibration [34,35], whereas the peak at 1630 cm^−1^ is attributed to the C==O stretching of the -CONH_2_ group (amide I) mode [30]. The shifts of the peaks from 1650 cm^−1^ and 1567 cm^−1^ of CS to 1630 and 1535 cm^−1^ (N-H bending amide II mode) of CS-NPs indicate an interaction between the NH_3_^+^ groups of chitosan and phosphate groups of TPP [36]. Additionally, the spectra for all CS-NPs show the asymmetric stretching mode of the C-O-C bridge of glucose-β-1-4 at 1154 cm^−1^ [35], a peak at 1064 cm^−1^ (C-O stretching) and at 1025 cm^−1^ (C-O bending). TPP incorporation in CS-NPs is also indicated by the presence of the 890 cm^−1^ peak [37].

Overall, our XRD, TEM, and FTIR results show consistency as to the degree of crystallinity and structural characteristics exhibited by our CS-NPs. Our discovery of additional structural characteristics of the CS-NPs, namely those associated with features located near 12° and 30° in the XRD spectra, is most likely attributable to the variability of the CS-related structures that have yet to be fully examined. Although there is generally a reduction in the degree of crystallinity of our CS-NPs synthesized using ionic gelation compared to that of the source CS, our novel results show that functionalization using the PEO-PPG copolymer clearly enables greater retention of crystallinity. The heterogeneous nature of the degree of crystallinity of the CS-NPs, as determined using HRTEM, indicates that ionic gelation may not be the most suitable method of synthesis if more structurally homogenous nanoparticles are desired. In this case, a better synthesis methodology may involve using a microemulsion (i.e., reverse micelle). An additional novel characteristic revealed from the HRTEM analysis is that for CS-NPs having a high degree of crystallinity, the ordered structure extends to the surface of the nanoparticles. This may be critical for the tunability and/or efficacy of surface functionalization with polymers and ligands, depending upon conditions where such properties may be predominantly governed by the surface structural characteristics of CS-NPs. Our next objective is to load our CS-NPs with known cancer treatment drugs and test their drug delivery potential in HeLa cell culture studies. The results of these studies will be published in the future.

## 4. Conclusions

This paper focuses on the structural, morphological, and molecular properties of CS-NPs synthesized via the ionic gelation method and specifically tailored for drug delivery purposes. CS-NPs were synthesized at concentrations ranging from 0.25% to 1.0% *w*/*v*. Furthermore, the 1.0% *w*/*v* CS-NPs were functionalized separately with PEO and with the PEO-PPG block copolymer. The average size of the CS-NPs, as determined using TEM imaging, was in the 11.3 to 14.8 nm range. XRD and HRTEM analyses revealed that the CS-NPs possess a semi-crystalline structure, with the degree of crystallinity reduced substantially compared to that of bulk CS. The CS concentration and functionalization with PEO were found to have no discernible effect on the degree of crystallinity of the CS-NPs. However, functionalization with PEO-PPG led to a significant increase in their degree of crystallinity. FTIR analysis confirmed the incorporation of TPP with CS, in all CS-NP samples, and an enhancement of hydrogen bonding in the CS-NPs functionalized with PEO-PPG. Our XRD results uncovered new CS-NP structural characteristics, which are potentially due to the variability of CS-related structures. HRTEM analysis revealed that the degree of crystallinity of CS-NPs is distributed heterogeneously, with the more crystalline nanoparticles trending toward a prismatic-shaped morphology. The less crystalline or amorphous nanoparticles trend toward a pseudo-spherical morphology. The findings provide valuable insights into potential pathways toward optimization of the structural characteristics of functionalized CS-NPs for enhanced efficacy and/or tunability for drug delivery.

## Figures and Tables

**Figure 1 bioengineering-11-00372-f001:**
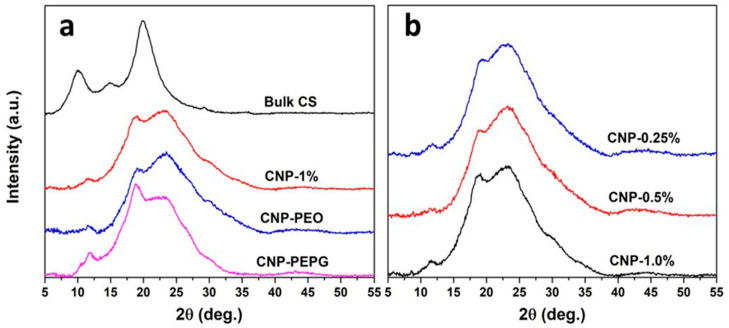
(**a**) XRD spectra plotted as intensity (a.u.) vs. detection angle 2θ and measured from the CS-NP samples without PEO and functionalized with PEO and with PEO+PPG, compared to that of bulk CS; (**b**) XRD spectra measured from the CS-NP samples as a function of CS concentration.

**Figure 2 bioengineering-11-00372-f002:**
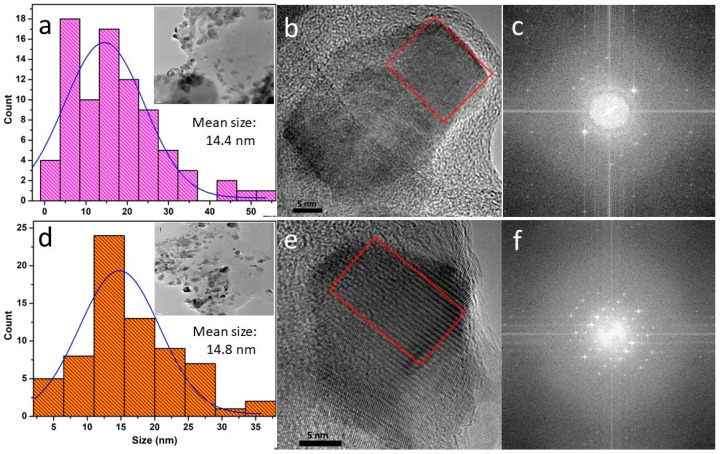
(**a**) Size distribution histogram with inset showing a TEM image, (**b**) a high-resolution TEM image, and (**c**) a FFT of the region delineated by the rectangle in (**b**) of sample CNP-1.0%; (**d**) size distribution histogram with inset displaying a corresponding TEM image, (**e**) a high-resolution TEM image and (**f**) a FFT of the region outlined by the rectangle in (**e**) of sample CNP-PEPG.

**Figure 3 bioengineering-11-00372-f003:**
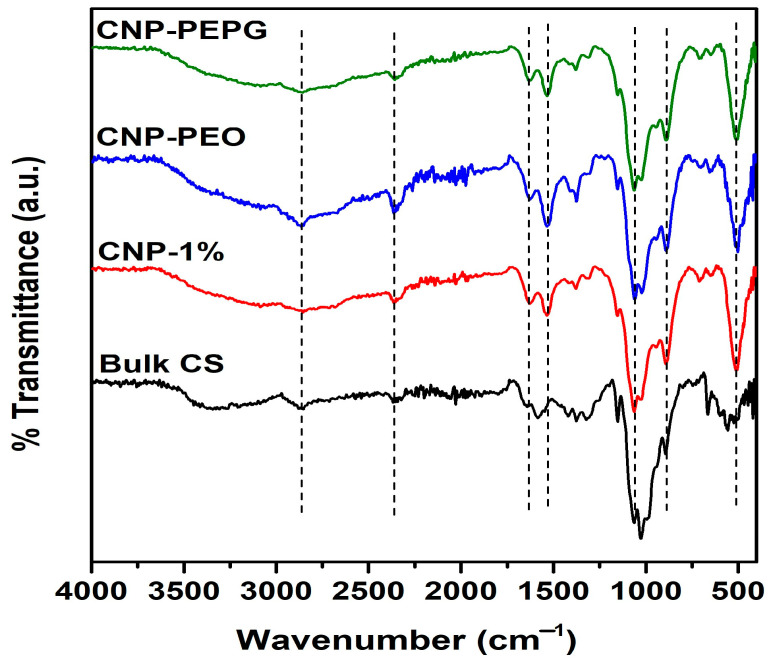
FTIR spectra measured from the samples of this study.

**Table 1 bioengineering-11-00372-t001:** CS-NP samples of this study and their mean size as determined from TEM measurements.

Sample	Description	Mean Size (nm)	FWHM (nm)
CNP-0.25%	0.25% *w*/*v* CS-NPs	-	-
CNP-0.5%	0.5% *w*/*v* CS-NPs	13.1	15.4
CNP-1.0%	1.0% *w*/*v* CS-NPs	14.4	22.7
CNP-PEO	1.0% *w*/*v* CS-NPs+PEO	11.3	13.0
CNP-PEPG	1.0% *w*/*v* CS-NPs+PEO+PPG	14.8	14.0

## Data Availability

Data are available by the authors on request.

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
