# Peer review of "On the Structural and Molecular Properties of PEO and PEO-PPG Functionalized Chitosan Nanoparticles for Drug Delivery"

_bioengineering, 2024, doi:10.3390/bioengineering11040372_

Round 1

Reviewer 1 Report

Comments and Suggestions for Authors

Although the paper is significant, there are several issues many missing parts should be addressed before the decision, and I recommend major revision.

  • The abstract is quite long. Consider shortening it to focus only on the key points and findings. Remove some of the background details.
  1. Introduction:
  • Provide some more context in the opening paragraph about why CS-NPs are being studied for drug delivery applications.
  • The introduction covers a lot of background on chitosan structure. Narrow the focus to only the most relevant details needed to understand this study.
  • Clearly state the motivation and objectives at the end of the introduction.

  1. Materials and Methods:
  • Consider moving Table 1 to the results section since it contains data.
  • In the characterization section, briefly explain the purpose of using each technique (XRD, TEM, FTIR) rather than just listing the instruments used.
  1. Results:
  • In Figure 1, use more descriptive axis labels (e.g. Intensity (a.u.) instead of just Intensity). Add unit labels in the caption.
  • Figure 2 is unclear without referring back to Table 1. Consider combining Table 1 into Figure 2 to better showcase the size data.
  • For FTIR results, highlight the 1-2 key peaks that indicate CS-NP formation/crystallinity rather than describing all observed peaks.
  1. Discussion:
  • Relate the results back to the introduction objectives. Focus the discussion on the most important findings regarding CS-NP crystallinity.
  1. Conclusion:
  • Summarize only the key conclusions, rather than repeating results. Relate back to the motivation and importance of this work.
  1. Writing:

  • Avoid long sentences throughout. Break up into shorter sentences for clarity.
  • Define all abbreviations upon first use.
  • Carefully proofread to fix typos, formatting issues, and improve flow.
Comments on the Quality of English Language

 Extensive editing of English language required

Author Response

Hello,

Thank you for your suggestion and comments. Please See the attachment below for the replies.

Sincerely,

Reviewer 2 Report

Comments and Suggestions for Authors

The general purpose of the work is not clearly defined:

The authors declare that "Chitosan nanoparticles are currently under vigorous investigation as suitable agents for in vivo delivery of drugs and genetic material."

However, the topic of what examples of the applications in biomedicine are supposed with the chitosan nanoparticles developed in the study and how they differ from existing ones is not disclosed.

The novelty of the developed particles or the idea of their application is not discussed either in the introduction or in the conclusions.

Comments on the Quality of English Language

Minor editing of English language required

Author Response

(The authors gave the same response as above.)

Round 2

Reviewer 1 Report

Comments and Suggestions for Authors

Accept

Comments on the Quality of English Language

Extensive editing of English language required

Reviewer 2 Report

Comments and Suggestions for Authors

the article has been significantly improved by the authors, but the novelty of the proposed development is still insufficiently emphasized,

also, it is not enough discussion in terms of the comparison of the results obtained with the results of other works (with literature data)

Comments on the Quality of English Language

Minor editing of English language required
